# Do Small Incisions Need Only Minimal Anesthesia?—Anesthetic Management in Laparoscopic and Robotic Surgery

**DOI:** 10.3390/jcm9124058

**Published:** 2020-12-15

**Authors:** Sebastian Hottenrott, Tobias Schlesinger, Philipp Helmer, Patrick Meybohm, Ibrahim Alkatout, Peter Kranke

**Affiliations:** 1Department of Anesthesiology and Critical Care, University Hospital Wuerzburg, 97080 Wuerzburg, Germany; hottenrott_s@ukw.de (S.H.); schlesinge_t@ukw.de (T.S.); helmer_p@ukw.de (P.H.); meybohm_p@ukw.de (P.M.); 2Department of Gynaecology and Obstetrics, Kiel School of Gynaecological Endoscopy, University Hospitals Schleswig-Holstein, 24105 Kiel, Germany; ibrahim.alkatout@uksh.de

**Keywords:** general anesthesia, anesthetics, perioperative care, minimally invasive surgery, laparoscopic surgery, robotic surgery

## Abstract

Laparoscopic techniques have established themselves as a major part of modern surgery. Their implementation in every surgical discipline has played a vital part in the reduction of perioperative morbidity and mortality. Precise robotic surgery, as an evolution of this, is shaping the present and future operating theatre that an anesthetist is facing. While incisions get smaller and the impact on the organism seems to dwindle, challenges for anesthetists do not lessen and could even become more demanding than in open procedures. This review focuses on the pathophysiological effects of contemporary laparoscopic and robotic procedures and summarizes anesthetic challenges and strategies for perioperative management.

## 1. Background

Since its introduction to the operating theatre, laparoscopic surgery has become a mainstay of surgical management. The evolution of minimally invasive techniques enabled laparoscopic surgery and has come a long way until today. This evolution ranges from Bozzini, who in 1805 tried to observe the urethra with a simple tube and candlelight [1], to the first use of pneumoperitoneum by Kelling in 1901, the first clinical implementations in diagnostics and treatment by gynecologists to the first laparoscopic cholecystectomy performed by the surgeon Phillipe Mouret in 1988 [1,2]. Remarkable advances in laparoscopic surgery have led to multiple benefits such as reduced blood loss, tissue trauma, postoperative morbidity and pain [3,4]. Many procedures now require a shorter hospital stay or can be performed on an ambulatory basis [5,6]. Furthermore, robotic surgery as the technical pinnacle of laparoscopic technique has rapidly evolved to be used in different surgical procedures and is a vital option for otherwise inoperable obese patients [7,8]. As the clinical implications for laparoscopic and robotic surgery broaden and the procedures get more complex, different challenges arise for the anesthesiologist. Here, we give an overview about the operative conditions, the physiological and pathophysiological changes and possible complications which have to be considered by the anesthetist. Additionally, we review the evidence for the anesthetic management of patients undergoing laparoscopic and robotic surgery.

## 2. Technique of Laparoscopic and Robotic Surgery

To successfully master potential challenges of laparoscopic and robotic surgery it is vital to understand the basic principles of their technique. The laparoscopic and robotic site is defined by the use of smaller and more precise incisions compared to their counterparts in open surgery, which leads to a reduction of tissue trauma [3]. To make room for optimal visualization of the targeted abdominal region pneumoperitoneum usually needs to be established, although other techniques, such as the “Abdominal lift” have been studied and applied to ensure visualization and accessibility [9]. Pneumoperitoneum is achieved by the insufflation of the abdominal cavity with gas [10]. Many different gases have been studied for induction of pneumoperitoneum, including room air, oxygen (O_2_), nitrous oxide (NO) and carbon dioxide (CO_2_). CO_2_ has been found to be the safest option, as its risk for gas embolism is lower than for O_2_ and room air due to higher resorption rate. There is increased risk for intra-abdominal combustion if NO is mixed with methane produced by the intestines [11,12]. Usually, laparoscopic access is done via a trocar or a needle through which the gas can be insufflated [13]. After pneumoperitoneum has been established, the surgeon is able to advance instruments via different trocar locations into the abdominal cavity. View of the operating site is generated via a camera trocar. To establish good sight of the targeted abdominal region, specific positioning of the patient is established. While many procedures can be done in supine position, different unphysiological positions like Trendelenburg, anti-Trendelenburg or lithotomy position are sometimes needed. As for standard laparoscopy, the surgeon and the operating team stand next to the patient. In modern robotic laparoscopy the surgeon can use telesurgery to control his instruments from a different location other than the operating table (e.g., DaVinci^®^). The advantages and disadvantages of laparoscopic and robotic surgery for the patient are summarized in Table 1.

## 3. Anesthetic Challenges

### 3.1. Hemodynamics

Many different factors of laparoscopic surgery influence the hemodynamic situation of the patient. Most notably, this is seen in the increase of intraabdominal pressure (IAP) due to the inflation with CO_2_ and positioning of the patient. With the initiation of pneumoperitoneum the IAP slowly rises and therefore pressure on the splanchnic veins is higher. This leads to an increase of preload followed by a higher blood pressure in the early phase of the laparoscopic procedure. Pain and neuro-humoral response to peritoneal stretching might also release catecholamines, which can lead to critical rises in blood pressure and tachyarrhythmia [15]. If IAP exceeds intravenous pressure in the splanchnic veins (≥15 mmHg), venous collapse is possible; this effect is reversed up to enhance inferior V. cava compression. Surgical manipulation and vasodilatation induced by anesthetic agents, release of prostaglandins (eventeration syndrome) and co-morbidities might add to hemodynamic instability. Changes in the patient’s position play an important role as well. Surgery in the upper abdominal quadrants like cholecystectomy needs anti-Trendelenburg positioning, which might lower Cardiac Index (CI) due to reduced venous return from the lower body parts. Procedures in the lower abdominal quadrants often need Trendelenburg positioning. In consequence of the elevated lower body, venous return increases, which raises CI, central venous pressure (CVP) and intracranial pressure (ICP) [15,16]. Depending on the phase and the technique of the procedure, the anesthetist should therefore be prepared for either critical hypertension or hypotension. Close monitoring of IAP is warranted with a target of ≤15 mmHg [17].

### 3.2. Respiratory Function

General anesthesia with progressive muscle relaxation already reduces residual capacity of the lung by up to 20% [18]. This effect is based on paralyzation of the diaphragm, which limits ventilation of basal lung segments and facilitates development of atelectasis. If pneumoperitoneum is applied and IAP increases in laparoscopic surgery, lung ventilation is increasingly compromised due to cranial movement of intraabdominal organs and rising pressure on the diaphragm. Trendelenburg position of the patient adds to this even further. Compliance of the lung and thorax is therefore impaired in the patient undergoing laparoscopic surgery and lung protective ventilation might be a challenge for the anesthetist, especially when iatrogenic hypercapnia must be compensated for by increased minute ventilation. Intraoperative impairment of lung ventilation and formation of atelectasis also have an impact on postoperative lung function [19]. Vital capacity and forced expiratory volume are found to be reduced in patients after laparoscopic surgery [20].

### 3.3. Renal and Hepatic Function

Abdominal blood flow inversely correlates to IAP. As IAP rises and exceeds arterial pressure of the abdominal arteries, perfusion of abdominal organs like kidney and liver will be impaired and venous drainage reduced. The reduction of renal blood flow causes a reduction of glomerular filtration with a decrease of urine secretion and retention of creatinine. Optimization of intravascular volume can help to mitigate this effect [21]. The anesthetist should therefore closely monitor the intravascular volume state, as well as perioperative diuresis and kidney function of the patient. If blood flow to the liver or venous drainage is reduced, hepatic dysfunction and a rise of liver enzymes can be seen [22]. In patients with normal liver function these effects are mostly self-limited and are not associated with any morbidity [23]. If there is preexisting impairment of liver function or the rise of IAP is severely prolonged, liver function should be of concern for the anesthetist.

### 3.4. CO_2_ Resorption

CO_2_ in the abdominal cavity as in pneumoperitoneum will be resorbed into the blood stream over time. Abdominal CO_2_ resorption depends on IAP, location of CO_2_ application and phase of the procedure [18]. There is evidence that extra-peritoneal insufflated CO_2_ as used in laparoscopic prostatectomy will be resorbed much faster than intraperitoneal CO_2_ [24]. At the start of the procedure CO_2_ resorption will rise with the induction of pneumoperitoneum, reach a steady state and might shortly rise again during the release of pneumoperitoneum at the end of the procedure due to the increase of venous return with falling IAP. Resorbed CO_2_ is then transported to the lungs and will be exhaled along with CO_2_ produced by the normal metabolism. As CO_2_ partial pressure is a determining factor of acid base homeostasis, adjustment of ventilation is required to prevent hypercapnia and acidosis. Minute ventilation has to be increased by up to 15% to match the respiratory demand generated by increased CO_2_ resorption in pneumoperitoneum [25]. The use of a valve-less trocar by the surgeons has been shown to improve respiratory mechanics in robot-assisted radical cystectomy and could help to ease ventilatory demands if prolonged pneumoperitoneum is required [26]. If ventilatory adjustment is not possible, the surgeons should be asked to reduce IAP or stop the CO_2_ flow for a few minutes. Otherwise, hypoventilation leads to hypercapnia and acidosis. Increased pCO_2_ and acidosis lead to vasodilatation of cerebral vessels, followed by an increase of cerebral blood flow and ICP. This can be fatal in patients with already elevated ICP. Acidosis leads to pulmonary vasoconstriction, which results in a higher pulmonary vascular resistance, with increased right-left shunt, which can lead to cardiac decompensation in patients with right heart failure. It is therefore vital to always monitor gas exchange, acid base homeostasis and end-tidal pCO_2_ (p_et_CO_2_) of the patient.

### 3.5. Positioning

As already described, laparoscopic and, even more, robotic surgery demand particular positioning to access the targeted abdominal region. Besides the impact on hemodynamics and lung function, patient positioning can be tremendously detrimental for the accessibility of the patient. Covering of the head and extremities by surgical drape and positioning of the patient with possible intraoperative changes might inhibit control of the airway, vascular access and monitoring devices of the patient. Close attention should also be paid to pressure-free positioning to prevent damage to skin, nerves and especially to the eyes. Anesthetists should plan patient positioning together with the surgeons and evaluate monitoring as well as options for airway and vascular access to anticipate potential intraoperative problems.

### 3.6. Intra- and Postoperative Complications

Many different complications can arise during anaesthesia of patients undergoing laparoscopic and robotic procedures. In Table 2 and Table 3 we provide an overview about incidence, pathophysiology, diagnosis and management of important intra- and postoperative complications.

## 4. Anesthetic Management

### 4.1. Patient Selection

After laparoscopic surgery was met with skepticism in its early days, it is now seen as a safe and viable surgical option to treat a broad array of indications. If feasible, it can be advantageous in patients with underlying cardiac and pulmonary diseases and can be safely performed on frail patients [10,41]. Reduced blood loss and tissue trauma, less pain and shorter hospital stays can make laparoscopy better in the longer term for the less fit patient [42]. However, intraoperative stress on the cardiopulmonary system, extreme positioning and occasionally prolonged operating times demand careful patient selection. For relative contraindications like significant cardiovascular and cerebrovascular disease, thorough preoperative evaluation and judgement in a team with the surgeon are key to enabling surgery with a good outcome. Clear contraindications have been narrowed down to only a few in adult patients. In patients with increased ICP, laparoscopic surgery should be avoided. Absorbed CO_2_ and increased IAP due to pneumoperitoneum and extreme positioning pose the risk for further increase of ICP. In children, the detrimental effects of pneumoperitoneum and positioning on hemodynamics and pulmonary function might be even more pronounced [18]. Therefore, more caution is warranted for young patients with underlying cardiovascular and pulmonary diseases.

### 4.2. Preoperative Considerations

Preoperative evaluation of the patient is the basis for successful anesthesia. It is composed of risk stratification based on the patient’s medical history and physical examination, including functional assessment and preoperative testing [42]. The anesthetist should check for cardiovascular, pulmonary and metabolic issues in the patient as these pose the highest risk to be aggravated by laparoscopic surgery. Treatment of underlying cardiopulmonary and metabolic diseases, as well as optimization of medication should then be carried out if possible. Careful planning of airway management is warranted, as limited accessibility, CO_2_ resorption, extreme positioning with increased IAP are challenges to overcome. Potential occurrence of facial or laryngeal edema might require a delay of extubation or the use of advanced techniques for airway management [43]. As intraoperative bleeding happens less often during laparoscopic procedures, it can be highly difficult for the surgeon to contain [44]. Adequate patient blood management is crucial in perioperative management of patients undergoing laparoscopic procedures [45]. It is important to avoid preoperative coagulopathy and anemia, which is an important predictor of perioperative morbidity and mortality itself [46,47,48,49]. Therefore, it is always worthwhile to screen for and treat anemia, but most important in laparoscopic procedures predicted to have a higher risk of blood loss of ≥500 mL or likelihood of transfusion of ≥10%. Laparoscopic tumor resection and vascular surgery typically belong in this category.

### 4.3. Choice of Technique

To induce and maintain anesthesia in laparoscopic surgery, total intravenous anesthesia (TIVA) with propofol or balanced anesthesia with use of inhalational agents present the options for general anesthesia. Each technique has different advantages and disadvantages. Use of propofol in TIVA lowers the risk of postoperative nausea and vomiting (PONV) and reduces pollution of the operating theatre and environment with anesthesia gas [50]. Propofol could also prevent an increase in intraocular pressure (IOP) after pneumoperitoneum and Trendelenburg positioning compared with sevoflurane [7,50]. Whereas balanced anesthesia has the benefits of easier monitoring and potentially safer supply of anesthesia if venous access is not in sight, e.g., if arms are attached to the body during the procedure, and, although controversially discussed even in cardiac surgery, the proposed cardio-protective effects of inhalational anesthesia [7,51]. Suppression of spinal reflexes is also more pronounced through volatile anesthetics such as sevoflurane compared to propofol [52,53]. As instruments in laparoscopic and especially robotic surgery cannot be retracted fast enough in response to sudden unpredictable movement of the patient, spinal reflexes should be sufficiently suppressed. General anesthesia with continuous analgesia through remifentanil might help to reduce the occurrence of these to a minimum. However, no superiority has been shown for either of these techniques in laparoscopic and robotic surgery and the final choice needs to be made by the anesthetist, depending on experience and setting. Neuraxial anesthesia could be another option, but is usually limited to lower abdominal and pelvic procedures. However, evidence shows an association with intraoperative pain referred to the shoulder, required anesthetic conversion in 3.4% of the cases and no respiratory benefits for patients with normal pulmonary function [54]. General anesthesia can be combined with epidural analgesia in extensive laparoscopic surgery with the benefit of lowered postoperative pain intensity through patient-controlled epidural anesthesia (PCEA). There is evidence that thoracic epidural analgesia provides additional benefits to patients with pulmonary risk factors undergoing laparoscopic surgery of the upper abdomen [55]. However, it is still unclear if the benefits of epidural anesthesia outweigh the risks for every patient. Therefore, careful individual benefit-to-risk consideration is needed. Airway management is of high importance to facilitate adequate ventilation under increased IAP in pneumoperitoneum with CO_2_ and extreme positioning. Here, higher ventilation pressures and prevention of aspiration are possible. While intubation represents the standard, the use of second generation supraglottic airway devices can also be considered in selected cases, e.g., in lower abdominal and pelvic procedures. Such devices offer optimized fit and the option to insert a stomach tube and might therefore be able to accomplish safe ventilation in laparoscopic procedures even in obese patients [56]. There is evidence from randomized double blind prospective studies for a reduction in PONV and postoperative throat pain and avoidance of hemodynamic changes occurring during endotracheal intubation, when laryngeal masks are used in comparison to endotracheal intubation [57,58]. Pressure-controlled ventilation and neuromuscular blockade have been proven to be optimal for lung-protective ventilation and sufficient oxygenation [10]. So far, there is some evidence that deep neuromuscular blockade can improve surgical conditions when compared to moderate degree of neuromuscular block, which might allow for lower IAP to be used [59]. However, there is still controversy as to whether this intervention is generally advisable or should only be limited to selected procedures or even be dedicated to difficult surgical situations [60,61]. Development of atelectasis should be controlled and treated by careful lung recruitment maneuvers and titration to adequate positive end-expiratory pressure (PEEP) [62]. Hemodynamic changes, especially intraoperative spikes in blood pressure cause the release of catecholamines and represent a big challenge to the anesthetist. To prepare for this event, continuous analgesia with remifentanil can be helpful. In addition, intravenous magnesium sulphate given before pneumoperitoneum can attenuate increases in arterial pressure during laparoscopic cholecystectomy [16].

### 4.4. Monitoring

As a standard in laparoscopic procedures, ECG, pulse oximetry and oscillometric blood pressure measurement should always be monitored. Due to induction of pneumoperitoneum, it is also of high importance to monitor capnography and neuromuscular blockade, ideally through the procedure, e.g., by using train-of-four (TOF) monitoring. Capnography allows timely adaption of ventilation to expiratory CO_2_. As the differences between p_a_CO_2_ and p_et_CO_2_ (normal range 2–5 mmHg) can vary depending on patient age, regular adjustment through arterial blood gas analysis is warranted [63]. Close monitoring of p_et_CO_2_, as well as p_a_CO_2_ helps to detect development of atelectasis and complications caused by laparoscopic procedures like gas embolus, pneumothorax and secondary one-sided intubation. Adequate muscular relaxation could be performed to optimize surgical conditions and allow for lower IAP in pneumoperitoneum. The anesthetist should therefore manage relaxation steered by TOF response along with intraabdominal manipulation. For deep neuromuscular blockade (NMB), post-tetanic count (PTC) needs to be monitored and evaluated. Deep NMB is typically considered to be present if the PTC is ~2, although the definitions vary to a great extent. If TIVA is chosen for anesthesia, its depth should be monitored through electroencephalogram (EEG) devices. Continuous assessment of anesthesia depth helps to avoid insufficient stages of anesthesia, which is especially dangerous in robotic surgery. Extended hemodynamic monitoring should be chosen for patients with cardiovascular risk factors as indications are the same as for patients undergoing open procedures. However, the anesthetist should anticipate limited accessibility after the start of the procedure and should implement adequate vascular access and monitoring beforehand, since any establishment of advanced hemodynamic monitoring or vascular access may only be realized with delay.

### 4.5. Perioperative Pain Management

Adequate perioperative pain management is one of the most important ways in which the anesthesiologist can improve patient comfort, which then allows earlier recovery and mobility and improved postoperative respiratory function. Laparoscopic surgery eases this task through smaller incisions and reduced tissue trauma. Therefore, analgesic requirements including opioids are lower compared to open surgery [10]. However, small incisions do not mean no pain because severe pain might not be expected. It has been shown that minor procedures, including laparoscopic approaches quite commonly result in unexpectedly high levels of postoperative pain [64]. It has been hypothesized that surgeries in which higher pain scores are anticipated, better adherence to evidence-based pain treatment recommendations and improved quality of care is provided [64]. As optimal pain management is still a controversially discussed topic, evidence-based recommendations are available in national guidelines by the European Society of Regional Anesthesia and Pain Management in the form of PROSPECT (PROcedure-SPECific postoperative pain managemenT) [65]. Looking at the local situation and performing a proper needs assessment may help to find adequate analgesic protocols. To secure basic analgesia for the postoperative period, pre- or intraoperative paracetamol and NSAID (e.g., ibuprofen) or COX-2 (e.g., parecoxib) selective inhibitors should be used. Preoperative dexamethasone is recommended for its analgesic and anti-emetic effects. Opioids are rescue medication if basic analgesia is not sufficient [65]. However, many laparoscopic procedures may not be well-tolerated without opioid analgesia even in the immediate postoperative period. Regional anesthesia with transversus abdominis plane (TAP) block may become a further option to reduce intraoperative opioid requirements [66]. Ultrasound guided transversus abdominis plane block has been shown to reduce morphine consumption and somatic pain after robotic partial nephrectomy [67]. However, beneficial effects still remain controversial as only limited evidence for TAP block in laparoscopic surgery is available [68]. Therefore, PROSPECT guidelines do not recommend TAP block for laparoscopic cholecystectomy, hysterectomy and sleeve gastrectomy so far [65]. Perioperative lidocaine infusion could also help to ease postoperative pain and enhance recovery but more evidence is needed [69]. As even the small port site incisions can be quite painful, infiltration with long-acting local anesthetics by the surgeon is recommended [66]. Shoulder tip pain is a common pain management challenge after laparoscopic procedures due to irritation of the diaphragm and the phrenic nerve through the pneumoperitoneum. This can be eased by thorough expulsion of the intraabdominal gas at the end of the surgery through aspiration and repeated lung recruitment maneuvers [37].

## 5. Special Patient Subgroups

### 5.1. Pregnant Women

It is not uncommon for pregnant women to need abdominal surgery that is not related to obstetrics. Mostly, these procedures are appendectomy or cholecystectomy, which are usually managed in laparoscopic technique [70]. There is sufficient evidence for the advantages of laparoscopic technique over open procedures in non-pregnant women [71]. Occasionally, laparoscopic appendectomy is still associated with an increased risk of miscarriage [48]. However, after adjusting for possible confounders laparoscopy shows no elevated fetal risk compared to an open procedure [72]. Thus, the decision for or against one method is therefore more dependent on the circumstances of the individual case and experience and should not be dependent on the technology itself. Importantly, the preference for one regional anesthesia as part of an appendectomy typically implies an open approach. Laparoscopic cholecystectomy should be the technique of choice in pregnant women [73]. Careful evaluation of the fetal and uterine state should be carried out pre- and postoperatively. To avoid aortocaval compression syndrome the patient might be positioned with lateral tilt, although the resulting benefit of this action is far from being clear and *IAP* should not surpass 15 mmHg [74]. The anesthetist should also be aware of potential harmful drug effects.

### 5.2. Children

Modern laparoscopic surgery and its advantages are also implemented in pediatric surgery. An anesthetist has to be careful as the physiological conditions in children differ significantly compared to adults. Because of the smaller abdominal cavity, increased IAP due to pneumoperitoneum and positioning has a much stronger impact on the cardiovascular system and the lungs. Vagal reflexes in response to mesenterial traction and abdominal distension occur more frequently and pose the biggest threat to pediatric hemodynamics [75]. Therefore, a lower IAP under 8 mmHg should be targeted and parasympatholytic medication should be prepared to be accessed fast. If Trendelenburg or anti-Trendelenburg positioning is needed, a tilt of 15° should not be exceeded [18]. Volume state should be monitored closely and adapted to tolerate the effects of pneumoperitoneum. Diuresis should not be used to monitor the volume state as infants and children can be anuric or oliguric in the first 45 min of the procedure. This is recompensated through increased diuresis up to 6 h after the procedure [75]. A correct and safe tube fixation should be implemented and consistent control is warranted as tube dislocation is more frequent in children due to abdominal distension after induction of pneumoperitoneum.

### 5.3. Obese Patients

In obese patients, special attention must be drawn to adjustment of ventilation since chest and lung compliance is known to be reduced per se [76]. The induction of a pneumoperitoneum as well as extreme patient positioning (Trendelenburg) can further deteriorate ventilation and contribute to atelectasis formation. Therefore, application of adequate PEEP and pulmonary recruitment maneuvers are especially vital in obese patients [77].

## 6. Conclusions

Laparoscopic and robotic surgery have become essential techniques in the modern operating theatre and offer various benefits for the patient. Smaller incisions with less tissue trauma through minimally invasive techniques are one of the main advantages. However, specific requirements, like extreme positioning and pneumoperitoneum are needed. These pose impactful risks on the physiology of the patient; various complications that will not be seen in open procedures might arise and could be difficult to manage. The anesthetist must carefully select which patients can undergo such procedures and must always be prepared for intraoperative difficulties, although absolute contraindications are rare. Small incisions also do not necessarily mean lower pain levels, as pain in patients undergoing laparoscopic procedures, like appendectomy is likely to be underestimated. Based on the aforementioned findings and statements the famous sayings, “Small incisions do not equal minimal anesthesia,” or “There may be minor surgery, but typically this is not associated with minor anesthesia,” are perfectly true.


**Key Points:**
➢Laparoscopic surgery offers impactful benefits through reduced tissue trauma and has become a mainstay of surgical technique.➢Robotic surgery has matured into a safe surgical option that may sometimes enable precise procedures on otherwise inoperable patients.➢Surgical requirements for laparoscopic and robotic technique, like pneumoperitoneum and extreme positioning induce undesirable pathophysiological changes on hemodynamics and pulmonary function and obstruct access to and visual control of the patient for the anesthetist.-Adequate airway management, hemodynamic monitoring, vascular access and careful patient positioning should be established preoperatively, as intraoperative adjustment might not be possible and procedures might take longer than open surgery.-High vigilance for hemodynamic and pulmonary changes, especially at the induction and release of pneumoperitoneum is warranted.➢Anesthesiologist and surgeon should approach the procedure as a team and communicate closely.➢Cautious patient selection and preoperative optimization of cardiovascular and pulmonary problems is key.➢Preoperative optimization of anemia and coagulation through Patient Blood Management is crucial, as intraoperative bleeding might be rare, but can be disastrous due to reduced accessibility.➢Adequate and evidence-based pain management is required for every laparoscopic surgery, as postoperative pain is often underestimated in procedures with less tissue trauma


## Figures and Tables

**Table 1 jcm-09-04058-t001:** Surgical advantages and disadvantages of laparoscopic and robotic surgery [3,4,5,6,7,8].

Advantages	Disadvantages
**Intraoperative:**	**Intraoperative:**
- reduced tissue trauma - reduced blood loss with lower need for transfusion of blood and coagulation factors [14] - visualization of the whole abdominal cavity	- higher cost - prolonged operating time - higher difficulty of technique, lack of haptic feedback - surgical control of bleeding complications might be delayed - certain surgical instruments not usable (e.g., automated cell salvage)
**Robotic surgery specific:**	**Robotic surgery specific:**
- articulation beyond normal manipulation - three-dimensional magnification and steadier camera position - filtering of tremor	- prolonged learning curve - needs careful positioning and more space
**Postoperative:**	**Postoperative:**
- better cosmetic results - reduced pain with lower need of analgesia - preserved lung function with shorter recovery time - reduced rate of wound infections - shortened in-hospital stay	- tissue damage and nerve injury due to positioning and prolonged operating time

**Table 2 jcm-09-04058-t002:** Intraoperative complications of laparoscopic and robotic surgery.

Complications	Vascular Injury	Gas Embolism	Secondary One-Sided Intubation	Volume Overload	Pneumothorax/Pneumo-Pericardium/Pneumo-Mediastinum	Carboxyhemo-Globinemia
Incidence	7–13.8% [27]	significant: 0.001–0.59% [28] subclinical: ~30% [29,30]	0.2% [31]	depending on the procedure [18]	0.01–0.4% [32]	0.03% [33]
Pathophysiology	Advancement of trocars and needles in major abdominal vessels.	Insufflation of CO_2_ into an injured vessel. Danger for embolism rises after *IAP* ≥ 10 mmHg. Portal veins are prone to embolism.	Cranial displacement of diaphragm and carina due to higher *IAP* from pneumoperitoneum and patient positioning.	Resorption of fluids through peritoneum potentially leading to hyponatremia, heart failure and lung edema.	CO_2_ spreading through the diaphragm along anatomical openings or surgical lesions.	Intraabdominal smoke formation and resorption over peritoneum [18].
Diagnosis	RR ^1^ ↓, HF ^2^ ↑, Hb ^3^~ (↓ later), lactate ↑ Communication with surgeons	RR ↓, HF~, arrhythmia SpO_2_ ^4^ ↓, p_AW_ ^5^~, etCO_2_ ^6^ ↑ (↓ later) auscultation (“machinery murmur”), transesophageal echocardiography (TEE)	RR~, HF~ SpO_2_ ↓, p_AW_ ↑, etCO_2_ ↓, Compliance ↓ auscultation	RR~, HF~, electrolyte disorder SpO_2_ ↓ auscultation, Point-of-care ultrasound (POCUS)	RR ↓, *HF* ↑ (↓ later) SpO_2_ ↓, p_AW_ ↑, etCO_2_ ↑ (↓), Compliance ↓ auscultation, POCUS	*RR*~, HF~ SpO_2_~, lactate ↑ COHb ^13^ ↑
Prophylaxis and Treatment	large bore i.v. access, close monitoring of hemodynamics and sufficient relaxation	immediate termination of pneumoperitoneum, hyperventilation, F_i_O_2_ ^7^ 1,0, (ECMO ^8^)	Endotracheal tube placement close to vocal cords, cautious monitoring of ventilation	cautious monitoring of hemodynamics, electrolytes and volume state (*CVP* ^9^, *PPV* ^10^)	cautious monitoring of ventilation and hemodynamics PEEP ^11^ > IAP ^12^	suctioning of developing smoke

Abbreviations: ↑—increase of, ↓—decrease of; ^1^ Blood pressure, ^2^ Heart rate, ^3^ Hemoglobin, ^4^ Peripheral capillary oxygen saturation, ^5^ Mean Airway Pressure, ^6^ Partial pressure of end-tidal CO2, ^7^ Fraction of inspiratory O2, ^8^ Extracorporeal membrane oxygenation, ^9^ Central venous pressure, ^10^ Pulse pressure variation, ^11^ Positive end-expiratory pressure, ^12^ Intraabdominal pressure, ^13^ Carboxyhemoglobin.

**Table 3 jcm-09-04058-t003:** Postoperative complications of laparoscopic and robotic surgery.

Complications	Cerebral Edema	Laryngeal Edema	Shoulder Tip Pain	Ocular Injury
Incidence	case reports [34,35]	2–22% [36]	up to 60% [37]	0.05–3% [38]
Pathophysiology	High *ICP* and capillary leak in consequence of increased *CVP* in Trendelenburg position with pneumoperitoneum. [34,35,38].	Increased *CVP* due to positioning and pneumoperitoneum [39].	Abdominal irritation of the diaphragm and phrenic nerve caused by high *IAP* and CO_2_-induced intraperitoneal acidosis [37].	Corneal abrasions and higher incidence of ischemic optic neuropathy due to incomplete eye closure, increased *ICP* and prolonged operating time [40].
Diagnosis	altered/depressed mental state	post-extubation stridor respiratory failure	shoulder pain 24 h up to 4 days after surgery, frequently on the same side of procedure	visual loss and ocular pain
Prophylaxis and Treatment	check for conjunctival edema, restrict angle of Trendelenburg position to 30°, restrictive fluid management [35]	check for conjunctival edema, cuff leak test [39],intraoperative corticosteroids, posture with head up position prior to extubation, prolonged observation and careful extubation	sufficient analgesia, brief Trendelenburg-positioning and repeated lung recruitment-maneuvers at the end of the procedure [37]	Protective eye coverings, limited time in steep Trendelenburg position, restrictive fluid management [40]

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
