# Peer review of "Do Small Incisions Need Only Minimal Anesthesia?—Anesthetic Management in Laparoscopic and Robotic Surgery"

_jcm, 2020, doi:10.3390/jcm9124058_

Round 1
Reviewer 1 Report
This is a good quality literature review designed to be read by non-anesthetists.
The authors could cite Philippe Mouret as the inventor of laparoscopic cholecystectomy.
Author Response
Dear Reviewer,
We would like to thank you for the suggestion to cite Phillipe Mouret as the inventor of laparoscopic cholecystectomy. The fact has been integrated into the Introduction as stated in Line 29-32: "From Bozzini, who in 1805 tried to observe the urethra with a simple tube and candlelight [1], over the first use of pneumoperitoneum by Kelling in 1901, the first clinical implementations in diagnostics and treatment by Gynaecologists to the first laparoscopic cholecystectomy performed by the surgeon Phillipe Mouret in 1988 [1,2]".
Kind regards,
S. Hottenrott
Reviewer 2 Report
Aim of the current manuscript was to review on the pathophysiologic effects of contemporary laparoscopic and robotic procedures, summarizing anaesthetic challenges and strategies. The topic is interesting and manuscript well built, howver there are a few critical aspects that authors need to review in order to improve overall quality of manuscript. Authors undervaluated the specific role of transversus abdominal plane block in the management of perioperative pain, please imprive this aspect (The Effects of Ultrasound-Guided Transversus Abdominis Plane Block on Acute and Chronic Postsurgical Pain After Robotic Partial Nephrectomy: A Prospective Randomized Clinical Trial - Pain Med. 2020 Feb 1;21(2):378-386. doi: 10.1093/pm/pnz214.) (J Clin Med. 2019 Oct 24;8(11):1774. doi: 10.3390/jcm8111774. Recovery from Anesthesia after Robotic-Assisted Radical Cystectomy: Two Different Reversals of Neuromuscular Blockade).
Moreover, there are a few surgical aspects that are able to improve anaesthetic management during minimally invasive surgery, please report it (Anesth Analg. 2017 Jun;124(6):1794-1801. doi: 10.1213/ANE.0000000000002027.A Prospective, Randomized, Clinical Trial on the Effects of a Valveless Trocar on Respiratory Mechanics During Robotic Radical Cystectomy: A Pilot Study)
Author Response
Dear Reviewer,
We are grateful thank you for your suggestions regarding the TAP block and the use of a valveless trocar. Both informations have been integrated into our review. We wrote about the use of valveless trocars in line 121-124. In lines 269-274 the evidence and advantages of TAP block is discussed.
Kind regards,
S. Hottenrott
Reviewer 3 Report
Good manuscript, well organized and pleasent to read. Adequate references, well illustrated tables. The sections of the text have been well treated and not verbose.
Author Response
Dear Reviewer, we would like to thank you for work of reviewing our article.
Kind regards,
S. Hottenrott
Reviewer 4 Report
The abstract should at least respond to the question raised by the title right no we don't have any clue which oblige the reader to search for a responde through the whole manuscript
Page 3 ligne 261 analgesia should be corrected
sub group for anesthesia : it would be nice to write few sentences for elderly frail patients and morbi obese patients which require specific care
it would be nice to mention the occurrence of hypercapnia above 60mmhg (ETCO2) is not wellcomed and if ventilatory adjustement can't be done, the anesthesia team could communicate with surgeons to decrease pressure or stop the gaz fro several minutrs
Author Response
Dear Reviewer,
We would like to thank your for your suggestions and corrections. As suggested we tried to answer the question raised by the title in the abstract in a more clear way as can be seen in lines 17-21. Communication with surgical team to help adjust ventilation by reducing IAP or stopping CO2-flow in case of hypercapnia has been added in lines 121-124. We added a subgroup for obese patients, where anaesthesiologic challenges are explained in lines 384-389 and mentioned that laparscopic surgery is safe for frail patients. We hope that these changes meet you requirements.
Kind regards,
S. Hottenrott